# Effects of Multi-Strain Probiotics and *Perilla frutescens* Seed Extract Supplementation Alone or Combined on Growth Performance, Antioxidant Indices, and Intestinal Health of Weaned Piglets

**DOI:** 10.3390/ani12172246

**Published:** 2022-08-30

**Authors:** Jian Li, Qianqian Zhang, Yong Zhuo, Zhengfeng Fang, Lianqiang Che, Shengyu Xu, Bin Feng, Yan Lin, Xuemei Jiang, Xilun Zhao, De Wu

**Affiliations:** 1Key Laboratory for Animal Disease-Resistance Nutrition of China Ministry of Education, Animal Nutrition Institute, Sichuan Agricultural University, 211 Huimin Road, Wenjiang District, Chengdu 611130, China; 2Department of Animal Resource and Science, Dankook University, Cheonan 31116, Korea

**Keywords:** multi-strain probiotics, *Perilla frutescens* seed extract, growth performance, intestinal health, antioxidation capacity, weaned piglet

## Abstract

**Simple Summary:**

Weaning piglets face stressors from changes in feed and environment, which affects their growth. To resolve this problem, we explored the separate effects of multi-strain probiotics and *Perilla frutescens* seed extract and their combined effect on weaning piglets. We found multi-strain probiotics or *Perilla frutescens* seed extract both improved the gain to feed ratio and antioxidant capacity. In addition, multi-strain probiotics improved jejunal villus height and the villus height/crypt depth ratio. *Perilla frutescens* seed extract improved ileal villus height. The interactive effects were observed in jejunal villus height and the villus height/crypt depth ratio, ileal villus height, and the gene expression of *IL-1β* and *mucin2* in the intestinal mucosa. This study shows that using either multi-strain probiotics or *Perilla frutescens* seed extract alone is more effective than their combined use in weaning piglets.

**Abstract:**

This study examined the effects of multi-strain probiotics (BL) and *Perilla frutescens* seed extract (PSE), alone or in combination, on weaning piglets. In total, 96 weaning piglets were allocated into four treatments: CON group (the basal diet), PSE group (basal diet + 1g/kg PSE), BL group (basal diet + 2 g/kg BL), and BL+PSE group (basal diet +1 g/kg PSE + 2 g/kg BL) according to a 2 × 2 factorial arrangement. The supplementation of BL or PSE improved the gain to feed ratio. Dietary BL reduced diarrhea occurrence and *Escherichia coli,* but increased *Lactobacillus* counts in the ileal digesta. Dietary PSE tended to increase *Lactobacillus* counts in the ileal digesta. Interactive effects were found in terms of ileal villus height, the gene expression of *IL-1β,* and malondialdehyde in the ileal mucosa. Dietary BL lowered malondialdehyde in the spleen, liver, and jejunal mucosa but increased the total antioxidant capacity (T-AOC) in the liver and ileum mucosa. The supplementation of PSE improved superoxide dismutase in serum and T-AOC in the liver, and reduced MDA in liver, spleen, and jejunum mucosa. Taken together, BL or PSE showed positive effects, improving growth and intestinal morphology and enhancing antioxidant capacity. However, their interaction showed no beneficial effects on the antioxidant indices and the intestinal morphology of weaned piglets.

## 1. Introduction

Probiotics are frequently used as additives in livestock feed as they are beneficial to the gut. The best-known probiotics are *Bacillus* and *Lactobacillus.* Acid-resistant *Bacillus strains* can colonize the intestine temporarily and modulate intestinal pH [1]. Notably, the bacteriocins secreted by *Bacillus* and *Lactobacillus* without toxicity can lyse cell walls and consume bacterial proton movement, leading to cell death [2], for example, resulting in an inhibitory effect on *Escherichia coli*, *Staphylococcus aureus*, *Salmonella typhimurium,* and *Clostridium perfringens* [3]. Among them, the beneficial effects of *Bacillus*-based strains on growth rate, gain to feed ratio, and the intestinal mucosa morphology of weaned piglets and preweaning calves were reported [4,5]. Zhang et al. [6] found that a *B. subtilis* mixture increased the abundance of beneficial microbiota, such as *Clostridium* and *Lactobacillus*, which, in turn, increased mucin production, strengthening the gut barrier. Simultaneously, *Lactobacillus sporogenes* (*L. sporogenes*) has a similar function as *Bacillus subtilis* (*B. subtilis*), residing in the gut and secreting digestive enzymes [7]. In particular, the abundance of *Lactobacillus* in the piglet gut is reduced when weaning, and supplementation with *Lactobacillus* strains is conducive to gut health by modulating gut microorganisms [8]. Giang et al. [9,10,11] reported that dietary supplementation of lactic acid bacteria and *B. subtilis* improved growth performance and lowered diarrhea occurrence in weaning piglets.

*Perilla**frutescens* seed extract (PSE) contains unsaturated fatty acids, vitamin E, protein, polyphenols, and flavonoids [12]. It shows a strong antioxidant capacity because of its phenolic acid and flavonoid contents, such as rosmarinic acid and luteolin, which can eliminate harmful oxygen ions and hydrogen peroxide [12,13,14]. In particular, luteolin was found to inhibit oral cariogenic streptococci [15]. In addition, alpha-linolenic acid in PSE, one kind of omega-3 in unsaturated fatty acids, is reported to inhibit inflammatory factors and prevent cardiovascular disease [16,17]. Reportedly, *Perilla* seed meal can improve the growth rate, feed intake, and gain to feed ratio of broilers, and is also effective in improving fiber digestibility [18,19]. Nevertheless, the application of PSE in weaning piglets has not been explored.

In the production of pigs, weaning stressors include nutrition, psychology, and environment, leading to villous atrophy, crypt hyperplasia [8], and the production of excessive free radicals. Consequently, the induced intestinal barrier increases membrane permeability, which is conducive to the invasion and proliferation of harmful bacteria [20]. Considering the benefit of multi-strain probiotics in stimulating growth, and the strong antioxidant properties of PSE, it was hypothesized that a combination of multi-strain probiotics (BL) and PSE could ease stress at weaning. Therefore, we aimed to explore the influences of BL and PSE used independently or together on the growth performance, antioxidant properties, and intestinal health of weaned piglets.

## 2. Materials and Methods

This study was conducted at the experimental base in Ya’an (Sichuan, China). Animal experiments were approved by the Animal Care and Use Committee of Sichuan Agricultural University (Sichuan, China, approval code:20181215), and conducted in accordance with animal protection laws.

### 2.1. Animals, Design, and Housing

In total, 96 Duroc × (Landrace × Yorkshire) weaned piglets (30 ± 1 days old) weighing 8.20 ± 0.29 kg were divided into 4 treatment groups according to a 2 × 2 factorial arrangement. Each treatment group consisted of 6 pens with 4 pigs (2 male and 2 female). Diets included: (1) control diet, basal diet; (2) PSE diet, basal diet + 1 g/kg PSE; (3) BL, basal diet + 2 g/kg BL; and (4) basal diet + 1 g/kg PSE+2 g/kg BL. The basal corn-soybean diet was formulated to meet the nutritional needs of NRC [21] for piglets (7–25 kg) and was free from any antibiotics, as shown in Table 1. Piglets were housed in the pens (2 m × 1.5 m × 0.8 m) equipped with a nipple drinker and feeder with free access to water and feed. The temperature was kept at 26–28 °C and relative humidity was kept at 55–65%.

The BL was provided by Zhihua Feed Science & Technology Co., Ltd. (Wuhan, China). It contained concentrations of 3.5 × 10^9^ CFU/g of *B. subtilis*, 1.75 × 10^9^ CFU/g *B. subtilis* var natto, and 1.75 × 10^9^ CFU/g *L. sporogenes*. They were all dormant spores. The PSE was provided by Chongqing Super Science & Technology Development Co., Ltd. It was extracted by supercritical CO_2_, containing ≥2 g/kg alpha-linolenic acid, ≥0.6 g/kg linoleic acid, and ≥0.14 g/kg flavonoids. In order to ensure the freshness of the feed and the activity of multi-strain probiotics, we formulated the feed once weekly and usually put it in cold storage.

### 2.2. Sample Collection

The individual fasted body weight (BW) was determined at 08:00 on days 8, 15, and 22. The feed allocation and refusals were recorded daily on a pen basis. Growth performance data included average daily gain (ADG), average daily feed intake (ADFI), and ADG/ADFI (G/F). Scores were given thorough observing signs of stool consistency, and the average stool score per pen was recorded twice daily. The fecal consistency score was determined using the 5-point judging method of Giang et al. [10]. Briefly, the feces consistency criteria were as follows: 0, hard bar or granular; 1, soft stools but shapeable; 2, unshaped; 3, watery stool. Scores of 2 and 3 were considered diarrhea.

On days 8, 15, and 22, blood was obtained by aseptic needle puncturing the front vena cava of one pig per pen (fasted for 6 h) whose body weight was closest to the mean of the treatment group and gathered into vacuum tubes containing heparin sodium (5 mL; Shandong Yongkang Medical Products Co., Ltd., Shandong, China). Serum was harvested by centrifuging blood samples at 4 °C, at 3000× *g* for 10 min and stored at −20 °C. When the feeding trial ended, six piglets in each treatment (whose individual body weight was closest to the mean of the treatment group) were euthanized with injection of pentobarbital sodium. Their intestine was stripped from the mesentery and immediately placed on ice. Approximately 2-cm length tissues cut from the middle of the duodenum, jejunum, and ileum were put into paraformaldehyde solution (4% *v*/*v*) for morphological analysis. Duodenal, jejunal, and ileal mucosa were collected for antioxidant analysis as described by Chen et al. [22]. The liver samples were taken from the largest lobe after removing the surface membrane and stored at −80 °C. Spleen tissues taken from the same place were frozen at −80 °C for antioxidant analysis. Individual digesta collected from the jejunum and ileum was stored in sterile Eppendorf tubes and rapidly stored at −80 °C rapidly until analysis of microbiota according to the description of Hu et al. [5].

### 2.3. Assay of Oxidant and Antioxidant Indices in Serum, Liver, and Spleen Tissues, and Small Intestine Mucosa

In terms of assay of oxidant and antioxidant indices, we used the method described by Chen et al. [22]. Briefly, tissue and mucosa samples (1.0 g) were homogenized in 5.0 mL of ice-cold phosphate buffer (pH 7.2–7.4) and centrifuged at 4000× *g* for 10 min. The obtained supernatants were stored at −20 °C until further analysis. The concentrations of malondialdehyde (MDA) and total antioxidant capacity (T-AOC), and activities of superoxide dismutase (SOD), glutathione peroxidase (GSH-Px), and catalase (CAT) in sera and supernatants were assayed with matching kits (Product Nos. A003-1-2, A015-2-1, A005-1-2, A001-3, and A007-1-1, Nanjing Institute of Jiancheng Biological Engineering, Nanjing, China). Inter- and intra-assay variations were kept below 15%. Protein concentrations in the tissues and mucosa (duodenum, jejunum, ileum, liver, and spleen) were detected using a BCA Protein Assay Kit (Beyotime Biotechnology Inc. Shanghai, China). All samples were analyzed in duplicate.

### 2.4. Small Intestine Morphology Analysis

As intestinal morphology plays a vital function in absorbing nutrients and changes after weaning [23], we evaluated the morphology of duodenum, jejunum, and ileum. As previously described [24], the fixed intestinal segments (jejunum and ileum) were rinsed using running water for 30 min and subsequently dehydrated with ethanol at varying concentrations. The tissues were cleared with xylene, embedded in wax, and sliced into 5 µm-thick slices using a Leica RM2235 microtome (Leica, Germany). Finally, these tissue slices were dewaxed and subjected to hematoxylin-eosin staining. For each well-oriented villus, 10 measurements were recorded for both villus height (VH) and crypt depth (CD) using Image Pro Plus 6.0 (Media Cybernetics, Inc., Bethesda, MD, USA). The average of these 10 measurements was used to represent the VH and CD for each tissue. The V/C ratio was calculated by dividing the VH value by the CD value.

### 2.5. Determination of Enzyme Activity in Digestive Species

As previously described [5]. Frozen digesta was weighed and diluted (1:9, *w*/*v*) with phosphate buffer before cells were disrupted using an ultrasonic homogenizer (Scientz-48L, Ningbo, China). The supernatant was collected after centrifugation (3500× *g*, 10 min, 4 °C) and used for detecting the activity of lipase, trypsin, and amylase using the corresponding kits with the catalog numbers of A054-2-1, A045-2-2, and C016-1-1 (Jiancheng Bioengineering Institute, Nanjing, China). Parallel determination was carried out for individual indices. Coefficients of inter- and intrasample variations were all controlled within 12%.

### 2.6. Total RNA Extraction and Quantitative Real-Time PCR 

To assess the effects of BL and PSE on the gut barrier genes and inflammatory factors, we measured the gut mucosal genes according to the method of Hu et al. [5]. The total RNAs of ileum mucosa (about 0.1 g) were extracted by Trizol (TaKaRa Biotechnology Co., Ltd., Dalian, China) following the manufacturer’s instructions. The extracted RNA was evaluated by a NanoDrop 2000 (Thermo Scientific, Wilmington, DE, USA) with the OD_260_/OD_280_ ratio in the range of 1.8 to 2.0, and quality was assessed via gel electrophoresis (1% *w*/*v* agarose). After dilution, 1 μg RNA was reverse transcribed using HiScript^®^ III RT SuperMix (+gDNA wiper, Vazyme Biotech Co., Ltd., Nanjing, China). The primers (Appendix A) for *mucin1*, *mucin2*, *zonula occludens-1* (*ZO-1*), *claudin1*, and *β-actin* were designed via Prime Premier 6.0 and synthesized by Sangon Biotech Co., Ltd. (Chengdu, China). Real-time quantitative PCR analysis was carried out using SYBR qPCR Master Mix (Vazyme Biotech Co., Ltd., Nanjing, China) and an ABI 7500 real-time PCR instrument QuantStudio 6 (Thermo, USA) with a reaction volume of 20 μL. For real-time PCR, samples underwent denaturation at 95 °C for 30 s, followed by 40 cycles of 95 °C for 10 s and 30 s at 60 °C. Individual samples were assayed in triplicate. Relative expression levels of target genes were calculated using the 2^−ΔΔCt^ method [25] by normalization to *β-actin*.

### 2.7. Microbial DNA Isolation and RT-qPCR 

The total bacterial DNA in ileal chyme (*n* = 6 for each treatment) was extracted with the Stool DNA Kit (Omega Bio-Tek Doraville, CA, USA) according to the manufacturer’s manual. Primers and probes (Appendix A) were designed via Primer Premier 5.0 (Premier Biosoft International, Palo Alto, CA, USA).

Microbial RT-qPCR analysis and measurement was carried out as described by Chen et al. [26]. The total number of bacteria was determined with CFX-96 RT-qPCR and SYBR^®^-qPCR Master Mix (Vazyme Biotech Co., Ltd., Nanjing, China) in a reaction volume of 25 μL, while the numbers of *Lactobacillus*, *Bifidobacterium*, *Bacillus,* and *E. coli* were determined with CFX-96 RT-qPCR with SuperReal PreMix (TIANGEN Biotechnology Co., Ltd., Tianjin, China) in a reaction volume of 20 μL. Microbial data were converted into log_10_- for analysis referring to the standard curve made by Chen et al. [26].

### 2.8. Statistical Analysis

Data were analyzed as a 2 × 2 factorial using the mixed procedure of SAS 9.4 (Cary, NC, USA). The BL, PSE, and interaction of BL and PSE were fixed effects, and the pigs were random effects. Duncan’s multiple comparison was used to separate means when interactive effects were significant. For serum parameters, diarrhea scores, ADG, ADFI, and G/F, the individual pen was regarded as an experimental unit. To analyze microbiota, relative gene expression, and antioxidant indicators of tissue, every euthanized pig was a statistical unit. Results were considered significant when *p* < 0.05 while 0.05 < *p* < 0.1 was considered to be a tendency.

## 3. Results

### 3.1. Growth Performance

Dietary PSE tended to reduce G/F (*p* = 0.063) but BL tended to improve G/F during days 8–14 (*p* = 0.068). Dietary BL increased the G/F of piglets (*p* = 0.034) from days 1 to 21, as shown in Table 2, while PSE increased the G/F (*p* = 0.006) from days 15 to 21. However, the combined supplementation of BL and PSE showed no interactive influence on the growth performance of piglets (*p* > 0.10).

### 3.2. Diarrhea Scores and Intestinal Microbes

As shown in Figure 1, BL supplementation reduced (*p* < 0.001) the diarrhea scores of piglets from days 8 to 14 and from days 1 to 21, whereas there was no interactive effect of BL and PSE on the diarrhea scores (*p* > 0.10). Dietary PSE tended to reduce diarrhea during days 1– 7 (*p* = 0.068). Piglets fed on diets containing BL had a higher (*p* = 0.006, Table 3) *Lactobacilli* count and a lower *E. coli* count in the ileum chyme (*p* = 0.008). The PSE diet tended to boost the proliferation of *Lactobacilli* (*p* = 0.057).

### 3.3. Intestinal Mucosa Morphology and Digestive Enzymes 

As shown in Table 4, dietary BL boosted the VH and V/C ratio in the jejunum (*p* < 0.05). The supplementation of PSE in the diet increased the VH in the ileum (*p* = 0.003) and tended to increase the ileal V/C ratio (*p* = 0.088). However, the interactive effects of BL and PSE were detrimental to the jejunal V/C ratio and ileal VH (*p* < 0.001). Additionally, there were interactive effects on VH and CD in the jejunum (*p* < 0.05), but neither increase nor decrease was found compared with other groups. In Table 5, PSE elevated the amylase activity in the jejunum (*p* = 0.011), while dietary BL induced higher trypsin activity in the ileum (*p* = 0.001). The interaction of BL and PSE decreased lipase activity in the jejunum (*p* = 0.049), but had the tendency to increase the trypsin activity in the ileum (*p* = 0.082).

### 3.4. Intestinal Barrier Gene Expression

As shown in Table 6, the interaction effect of BL and PSE downregulated the gene expression of *IL-1β* (*p* < 0.01) and tended to downregulate the gene expression of *ZO-1,* but upregulate that of *Claudin-1* (*p* < 0.10). Although there was an interaction in mucin2 (*p* = 0.036), neither an increase nor decrease was found compared with other groups. Supplementary BL significantly downregulated the gene expression of *IL-1β* (*p* < 0.01) but tended to upregulate *mucin2* gene expression (*p* = 0.053). PSE supplementation downregulated the expression of *IL-1β,* but upregulated *mucin2* gene expression (*p* < 0.05).

### 3.5. Oxidation and Antioxidant Indices in Serum, Liver, Spleen, and Small Intestine

In Table 7, PSE improved (*p =* 0.001) SOD activity on day 21, decreased the MDA concentrations in the serum on day 7(*p =* 0.011), and tended to decrease MDA concentration on day 14 (*p* = 0.094). The interactive effect of BL and PSE tended to increase SOD on day 21 and GSH-Px activity on day 14 (*p* < 0.10). There was an interaction in MDA concentration on day 7 (*p* = 0.046). The dietary BL had the tendency to reduce the MDA concentration on day 14 but tended to increase the GSH-Px concentration on day 14 (0.05 < *p* < 0.1). Piglets fed BL had higher T-AOC concentrations in the liver (*p* = 0.016, Table 8) and lower MDA concentrations in the spleen (*p* = 0.003), whereas piglets fed on the PSE diet had higher T-AOC concentrations in the liver and lower MDA concentrations in the liver and spleen (*p* < 0.01). Additionally, in Table 9, PSE lowered MDA concentration in the jejunal mucosa (*p* = 0.024, Table 9), while BL increased T-AOC concentrations in the ileal mucosa (*p* = 0.019) and tended to reduce MDA concentration in the jejunal mucosa (*p* = 0.090). An interactive effect was observed in the MDA concentration in the jejunal mucosa, in which the MDA was higher in the +PSE+BL group compared with the +PSE-BL group, but lower compared with the -BL-PSE group (*p* < 0.05).

## 4. Discussion

Weaning stress results in diarrhea, oxidant stress, poor immune function, and growth arrest [26]. Functional probiotics and plant extracts have been reported to ease these symptoms when weaning [7,26,27,28]. Therefore, it was necessary to explore the effects of BL and PSE alone and combined on weaned piglets.

### 4.1. Growth Performance

This study revealed that BL supplementation improved G/F of weaned piglets, but had no effect on ADFI during days 8–14 and 1–21. This result was in agreement with similar studies of *Bacillus* and *Lactobacillus* probiotics in weaned piglets [8,29,30]. As previously reported, *Bacillus* spp.-based probiotics can stimulate the activity of non-starch polysaccharide degrading enzymes and proteases, and augments nutrient digestibility [3]. Therefore, we guessed that the improved trypsin activity by BL may stimulate the decomposition of proteins and then increase nutrient absorption. Sequentially, the tendency improvement of ADG by BL may be contacted with lower diarrhea score during days 8–14, as probiotics can effectively prevent diarrhea in weaned piglets [31]. Likewise, PSE supplementation also improved the G/F from day 15 to day 21. Plant extracts containing essential oils, flavonoids, and hydroxybenzene can improve growth performance [32], which may result from improved nutrient digestibility. Reportedly, alpha-linolenic acid and alpha-linoleic acid in the PSE are easily absorbed due to the high digestibility of fat and protein [33]. In the present study, improved amylase activity by PSE was more conducive to decomposing starch to easily absorb maltose and glucose [34]. PSE tended to reduce diarrhea scores during days 1–7. However, PSE tended to reduce G/F during days 8-14 under the premise that there were no effects on the diarrhea score. As there was no research of PES on piglets, the most likely explanation was that in the former, the piglets had a maladjustment for 1 g/kg PSE.

The feed conversion ratio is a vital parameter in assessing the economic efficiency of animal husbandry. In this study, BL or PSE were shown to boost growth performance. This may be due to better gut characteristics, including healthy and stable histomorphology, balanced intestinal microflora, and better intestinal barrier integrity.

### 4.2. Diarrhea Scores and Microbes

The present study revealed that piglets fed multi-strain probiotics had lower diarrhea scores. The results are in agreement with the beneficial effects of *Bacillus spores* and *Lactobacillus* supplementation on reducing the incidence of diarrhea in weaned piglets [35,36]. In the report by Du et al. [36], *Bacillus WS-1* prevented diarrhea and deaths of newborn piglets caused by *E. coli* injections, as *Bacillus strains* are well known to secrete surfactin, which acts against pathogenic bacteria. In the present study, the decreased diarrhea scores in BL-fed piglets were probably associated with the reduced *E. coli* counts and the increased *Lactobacillus* counts. The modulation of intestinal microorganisms by oral probiotics has been proven effective to reduce harmful *E. coli* counts and enrich healthy bacteria [30,37]. Reportedly, PSE can inhibit *E. coli* proliferation in vitro [38], whereas it showed no role in adjusting *E. coli* numbers in vivo in this study, but tended to boost *Lactobacillus* counts in the late stage of the trial. However, the latest report revealed that the *Perilla* extract had no inhibition on *E. coli* or *Salmonella* in vitro [39]. On the one hand, the *Perilla* variety, harvest season, and planting environment will have impacts on its composition [39]. On the other hand, when entering the body, PSE may be digested by digestive enzymes and absorbed by intestinal epithelial cells, so they cannot play a role in the hindgut.

### 4.3. Intestinal Mucosa Morphology

Weaning piglets are unsuitable for the abrupt changes in diet and line -haul with empty, resulting in reduced VH and increased CD [23,40]. In physiology, intestinal histomorphology is vital in assessing intestinal health and digestive function. We found that BL increased VH in the jejunum and ileum and increased the jejunal V/C ratio, which was in agreement with the results of Sayan et al. [41], who found that sucking piglets orally administered with *L. salivarius* had higher VH and V/C values in the jejunum and ileum. Meanwhile, piglets fed on diets containing PSE had numerically higher VH and V/C ratios in the ileum. Similar results were also reported by Wang et al. [29] with respect to the effects of herbs on weaning piglets. Plant extracts can stimulate the reconstruction of intestinal microvilli by activating the *cytokeratin* gene [42], possibly explaining the effects of PSE on morphology. The VH reflects the epithelial turnover speed and is related to the activation of cell mitosis [43], showing the protective effects of BL and PSE on epithelial mucosa. *B. subtilis* could prompt the development of wave-like villus arrays that increase the absorption area and prolong residence time in the digestive tract [44], which may explain the higher G/F in the BL groups.

### 4.4. Gut Barrier Gene Expression 

Weaning induces intestinal epithelial barrier damage, resulting in increased intestinal permeability, the invasion of intestinal pathogens, and decreased nutrient digestibility [44,45]. Therefore, an intact mucosal barrier is important to ensure nutrient absorption. This barrier is established by tight junction proteins, including *claudin* and *occludin* and intracellular linker proteins, such as *ZOs* [45]. In the current study, PSE upregulated the gene expression of *mucin2* but downregulated that of *IL-1β*. Phenolic acid and flavonoid constituents have a protective function on the gene expression of *ZO-1* and *occludin* because it is a potent inhibitor of protein tyrosine kinases that prevents tight junction protein from being phosphorylated [46]. Mucin secreted by goblet cells is regarded as a contributor to regulating intestinal inflammation by protecting against bacterial and parasitic invasions [45]. The high expression of the pro-inflammatory cytokine *IL-1β* disturbs tight junctions [29]. Interestingly, BL also decreased *IL-1β* gene expression. Strangely, PSE tended to improve the gene expression of TNF-α, the pro-inflammatory cytokines. The effects of PSE on gut tight junction proteins in weaning piglets have not been explored, and merit more attention in future studies. 

### 4.5. Antioxidant Capacity

Oxidant stress results in weaker growth performance and lower nutrient digestibility in weaned piglets [47]. Changes in diet, transport, and vaccine injection are major stressors, leading to the excessive production of free radicals that exceeds the removal rate of the antioxidant system, which leads to inflammation and damages the intestinal barrier. The levels of SOD, CAT, and GSH-Px are important antioxidant enzymes, and T-AOC in the body reflects the body’s ability to eliminate oxygen radicals. SOD converts reactive oxygen species into hydrogen peroxide; next, CAT and GSH-Px play roles in degrading unstable hydrogen peroxide to water and oxygen [22,48]. Another index is MDA, one of the final products of lipid peroxidation as well as an excellent oxidative stress marker [48]. In the present study, BL tended to augment GSH-Px and increased T-AOC capacity in the liver and ileum and reduced the MDA concentrations in the spleen and jejunum. Numerous studies have shown that *Bacillus* spp.-based probiotics improved T-AOC of sows and weaned lambs [48,49]. Probiotics enhanced the antioxidant capacity of animals by utilizing their antioxidant enzyme system of SOD, CAT, and NADH-oxidase or stimulating the host’s antioxidant system as well as probiotics-mediated antioxidant signaling pathways [50]. PSE induced higher SOD activity in serum and higher T-AOC capacity in the liver and decreased the MDA concentrations in the liver, spleen, and jejunum mucosa. *Perilla* extract consists of natural antioxidants, such as apigenin, citral, luteolin, perillaldehyde, quercetin, and rosmarinic acid, which chelate metal ions that produce free radicals and inhibit some enzymes involved in radical generation, due to special benzenoid rings and phenolic hydroxyl groups [51,52]. The results were linked to reports that plant extracts containing polyphenols and flavonoids strengthened CAT activity and the gene expression of *SOD1* and decreased MDA concentrate [53]. The enhancive antioxidant indices may reflect the protective effects of BL and PSE on the spleen, which involves the immune system; the liver, which participates in detoxification and metabolism [54]; and the intestinal mucosa, which maintains the intestinal barrier [55].

### 4.6. Interactive Effect

Interestingly, when piglet diets were supplemented with both BL and PSE, antagonistic effects on ileal VH and the jejunal V/C ratio were observed. Analogously, Zangeronimo et al. [56] certified that 1 g/kg plant extract mixed with 3 g/kg probiotics in the diet of weaned piglets decreased the VH compared with 0.5g/kg plant extract and 3g/kg of probiotics. The lack of increase in *E. coli* count in this period excluded the possibility of damage by microorganisms, and litter deeper CD predicted the possibility of intestinal inflammation [57], as proved by higher MDA concentrations in the jejunum and serum when BL and PSE were used together in this study. Maybe the unaffected growth performance was due to the above-mentioned and the decline in lipase activity. However, there was a contradiction between the downregulation of *IL-1β* expression, the upward trend of *Claudin-1,* the downward trend of *ZO-1,* and the increasing tendency in GSH-Px and SOD in serum by their interaction. These conflicting indicators are currently unexplained. Studies have found that PSE has a great antibacterial effect in vitro [58], but we are not sure of the specific action in the body.

## 5. Conclusions

This study provides the first evidence that PSE administration to weaned piglets increases G/F, strengthens the antioxidant capacity of the serum, liver, and jejunal mucosa, and ameliorates ileal morphology and barrier function. Feeding BL promoted G/F of weaned piglets, which may be associated with an enhanced antioxidant capacity in the liver, spleen, and ileal mucosa, and an improved jejunal morphology, and modulated the counts of *E. coli* and *Lactobacillus.* However, the combination of BL and PSE had no effect on growth performance, possibly because of the antagonistic effect on antioxidant capacity and intestinal morphology, but this remains to be determined.

## Figures and Tables

**Figure 1 animals-12-02246-f001:**
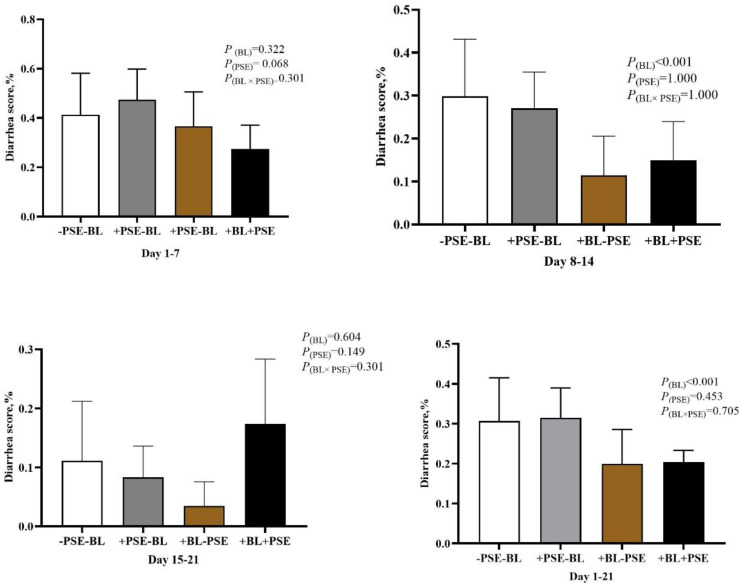
Effect of BL and PSE on the diarrhea rate of weaned piglets. Values are presented as means, *n* = 6. BL, multi-strain probiotics; PSE, *Perilla frutescens* seed extract; −PSE−BL, basal diet; +PSE−BL, basal diet + 1 g/kg *Perilla frutescens* seed extract; +BL−PSE, basal diet + 2 g/kg multi-strain probiotics; +BL + PSE, basal diet+2 g/kg multi-strain probiotics + 1 g/kg *Perilla frutescens* seed extract.

**Table 1 animals-12-02246-t001:** Composition and nutrient levels of basal diet (air-dry basis).

Ingredients (%)	Content	
	Day 1–14	Day 15–21
Corn	35.92	47.53
Extruded corn	18.00	15.00
Peeled soybean meal	12.00	17.00
Extruded soybean meal	12.00	8.00
Fish meal (67% CP)	4.00	3.00
Porcine plasma protein powder (70% CP)	3.00	0.00
Low protein whey powder	10.00	5.00
Soy bean oil	2.20	1.60
CaHPO_3_	0.80	0.70
Limestone	0.60	0.60
Sodium chloride	0.30	0.30
*L*-Lys. HCI (98%)	0.40	0.48
*DL*-Met (99%)	0.12	0.10
*L*-Thr (98.5%)	0.15	0.16
*L*-Trp (98%)	0.01	0.03
Choline chloride	0.10	0.10
Vitamin/trace element Premix ^1^	0.40	0.00
Vitamin/trace element Premix ^2^	0.00	0.40
Total	100	100
Nutrient composition		
Digestible energy, (MJ/kg)	14.82	14.64
Crude protein, %	20.77	19.15
Calcium, %	0.81	0.70
Available phosphorus, %	0.37	0.32
Calculated standardized ileal digestible, %		
SID-Lys, %	1.46	1.35
SID-Met, %	0.36	0.31
SID-Thr, %	0.96	0.80
SID-Trp, %	0.20	0.22

^1^ The vitamin–mineral premix provided the following per kilogram of basal diet: Zn (ZnSO_4_·H_2_O), 165 mg; Mn (MnSO_4_·H_2_O), 10 mg; Fe (FeSO_4_·H_2_O), 165 mg; Cu (CuSO_4_·5H_2_O), 200 mg; I (KI), 0.3 mg; Se (Na_2_SeO_3_), 0.3 mg; VA, 14,000 IU; VD_3_, 4000 IU; VE, 30 IU; VK_3_, 4.7 mg; VB_1_, 4 mg; VB_2_, 10 mg; VB_6_, 6 mg; VB_12_, 0.04 mg; niacin, 40 mg; pantothenic acid. ^2^ The vitamin–mineral premix provided the following per kilogram of basal diet: Zn (ZnSO_4_·H_2_O), 100 mg; Mn (MnSO_4_·H_2_O), 5 mg; Fe (FeSO_4_·H_2_O), 100 mg; Cu (CuSO_4_·5H_2_O), 20 mg; I (KI), 0.15 mg; Se, 0.25 mg; VA, 14,000 IU; VD_3_, 4000 IU; VE, 30 IU; VK_3_, 4.7 mg; VB_1_, 4 mg; VB_2_, 10 mg; VB_6_, 6 mg; VB_12_, 0.04 mg; niacin, 40 mg; pantothenic acid, 20 mg; folic acid, 2 mg; biotin, 0.16 mg.

**Table 2 animals-12-02246-t002:** Effects of BL and PSE supplementation on growth performance of weaned piglets ^1^.

Items ^2^	PSE × BL				Pooled SEM	*p*-Value		
	−PSE	−PSE	+PSE	+PSE		PSE	BL	Interaction
	−BL	+BL	−BL	+BL				
Body weight, kg								
Day 0	8.20	8.21	8.21	8.22	0.29	0.973	0.986	1.000
Day 7	9.74	9.85	9.80	9.80	0.40	0.990	0.899	0.893
Day 14	11.72	12.08	11.81	11.82	0.54	0.879	0.745	0.759
Day 21	14.02	14.49	14.21	14.31	0.69	0.999	0.686	0.784
Average daily gain, g/d								
Day 1–7	220	226	226	225	17	0.670	0.960	0.659
Day 8–14	283	318	287	289	22	0.410	0.574	0.456
Day 15–21	328	346	343	355	22	0.513	0.596	0.887
Day 1–21	277	300	286	290	20	0.507	0.979	0.647
Average daily gain, g/d								
Day 1–7	335	351	348	346	24	0.869	0.770	0.719
Day 8–14	460	485	469	471	35	0.941	0.696	0.753
Day 15–21	553	577	557	561	43	0.892	0.743	0.829
Day 1–21	433	471	455	462	24	0.805	0.364	0.547
Gain/feed								
Day 1–7	0.651	0.665	0.651	0.651	0.014	0.614	0.589	0.589
Day 8–14	0.616	0.659	0.612	0.616	0.012	0.063	0.068	0.123
Day 15–21	0.595	0.622	0.622	0.635	0.009	0.006	0.294	0.727
Day 1–21	0.617	0.637	0.625	0.632	0.006	0.807	0.034	0.273

^1^ Values were presented as means ± pooled SEM, *n* = 6. ^2^ BL, multi-strain probiotics; PSE, *Perilla frutescens* seed extract; −PSE−BL, basal diet; +PSE−BL, basal diet + 1 g/kg *Perilla frutescens* seed extract; +BL−PSE, basal diet + 2 g/kg multi-strain probiotics; +BL + PSE, basal diet+2 g/kg multi-strain probiotics + 1 g/kg *Perilla frutescens* seed extract.

**Table 3 animals-12-02246-t003:** Effects of BL and PSE on microbes in the ileal digesta of weaned piglets (10 copies/g of wet digesta) ^1^.

Items ^2^	PSE × BL				Pooled SEM	*p*-Value		
	−PSE	−PSE	+PSE	+PSE		PSE	BL	Interaction
	−BL	+BL	−BL	+BL				
Ileum digesta								
Total bacteria	10.46	10.66	10.50	10.43	0.10	0.323	0.498	0.186
*Bacillus*	8.59	8.80	8.57	8.65	0.09	0.340	0.112	0.469
*Lactobacillus*	5.89	6.22	6.14	6.31	0.08	0.057	0.006	0.344
*E. coli*	6.94	6.32	6.64	6.28	0.13	0.194	0.008	0.309
*Bifidobacterium*	7.42	7.26	6.90	7.18	0.30	0.315	0.871	0.507

^1^ Values were presented as means ± pooled SEM, *n* = 6. ^2^ BL, multi-strain probiotics; PSE, *Perilla frutescens* seed extract; −PSE−BL, basal diet; +PSE−BL, basal diet + 1 g/kg *Perilla frutescens* seed extract; +BL−PSE, basal diet + 2 g/kg multi-strain probiotics; +BL + PSE, basal diet+2 g/kg multi-strain probiotics + 1 g/kg *Perilla frutescens* seed extract.

**Table 4 animals-12-02246-t004:** Effects of BL and PSE on intestinal mucosa morphology of weaned piglets ^1^.

Items ^2^	PSE × BL				Pooled SEM	*p*-Value		
	−PSE	−PSE	+PSE	+PSE		PSE	BL	Interaction
	−BL	+BL	−BL	+BL				
Duodenum								
VH, μm	376	379	357	364	21	0.408	0.813	0.919
CD, μm	276	289	277	276	17	0.735	0.734	0.702
V/C ratio	1.37	1.32	1.29	1.32	0.03	0.207	0.668	0.190
Jejunum								
VH, μm	288 ^B^	385 ^A^	333 ^AB^	324 ^AB^	18	0.643	0.026	0.007
CD, μm	221 ^A^	181 ^B^	176 ^B^	203 ^AB^	10	0.252	0.523	0.002
V/C ratio	1.32 ^C^	2.14 ^A^	1.93 ^AB^	1.58 ^BC^	0.11	0.808	0.041	<0.001
Ileum								
VH, μm	303 ^C^	325 ^B^	342 ^A^	310 ^C^	4	0.003	0.173	<0.001
CD, μm	153	156	157	144	5	0.433	0.338	0.136
V/C ratio	2.00	2.09	2.18	2.16	0.07	0.088	0.592	0.447

^1^ Values were presented as means ± pooled SEM, *n* = 6. ^2^ BL, multi-strain probiotics; PSE, *Perilla frutescens* seed extract; −PSE−BL, basal diet; +PSE−BL, basal diet + 1 g/kg *Perilla frutescens* seed extract; +BL−PSE, basal diet + 2 g/kg multi-strain probiotics; +BL + PSE, basal diet+2 g/kg multi-strain probiotics + 1 g/kg *Perilla frutescens* seed extract. ^A,B,C^ Different letters on the shoulders of peers indicate significant differences (*p* < 0.05).

**Table 5 animals-12-02246-t005:** Effects of BL and PSE on digestive enzymes in the jejunum and ileum ^1^.

Items ^2^	PSE ×BL				Pooled SEM	*p*-Value		
	−PSE	−PSE	+PSE	+PSE		PSE	BL	Interaction
	−BL	+BL	−BL	+BL				
Jejunum								
Amylase, U/mg protein	1.53	1.54	1.63	1.59	0.03	0.011	0.615	0.394
Trypsin, U/mg protein	47.91	48.77	48.78	48.50	0.75	0.699	0.698	0.463
Lipase, U/mg protein	18.35 ^B^	19.68 ^AB^	20.32 ^A^	19.42 ^B^	0.53	0.124	0.691	0.049
Ileum								
Amylase, U/mg protein	1.24	1.17	1.24	1.18	0.04	0.983	0.114	0.983
Trypsin, U/mg protein	17.87	20.83	18.63	19.67	0.53	0.714	0.001	0.082
Lipase, U/mg protein	18.90	18.81	19.76	18.61	0.76	0.668	0.423	0.498

^1^ Values were presented as means ± pooled SEM, *n* = 6. ^2^ BL, multi-strain probiotics; PSE, *Perilla frutescens* seed extract; −PSE−BL, basal diet; +PSE−BL, basal diet + 1 g/kg *Perilla frutescens* seed extract; +BL−PSE, basal diet + 2 g/kg multi-strain probiotics; +BL + PSE, basal diet+2 g/kg multi-strain probiotics + 1 g/kg *Perilla frutescens* seed extract. ^A,B^ Different letters on the shoulders of peers indicate significant differences (*p* < 0.05).

**Table 6 animals-12-02246-t006:** Effects of BL and PSE on intestinal inflammatory factors and barrier gene expression in weaned piglets ^1^.

Items ^2^	PSE × BL				*p*-Value		
	−PSE	−PSE	+PSE	+PSE	PSE	BL	Interaction
	−BL	+BL	−BL	+BL			
IL-1β	1.00 ± 0.09 ^A^	0.58 ± 0.02 ^B^	0.56 ± 0.06 ^B^	0.56 ± 0.02 ^B^	<0.001	0.001	0.001
TNF-α	1.00 ± 0.13	1.39 ± 0.22	1.36 ± 0.11	1.43 ± 0.13	0.092	0.794	0.295
ZO-1	1.00 ± 0.07	1.31 ± 0.18	1.34 ± 0.11	1.18 ± 0.12	0.429	0.550	0.076
Claudin-1	1.00 ± 0.17	1.56 ± 0.16	1.45 ± 0.13	1.48 ± 0.06	0.201	0.046	0.064
Mucin1	1.00 ± 0.09	1.03 ± 0.07	1.04 ± 0.10	1.08 ± 0.07	0.590	0.670	0.938
Mucin2	1.00 ± 0.22 ^B^	1.58 ± 0.05 ^AB^	1.68 ± 0.12 ^A^	1.63 ± 0.07 ^AB^	0.014	0.053	0.036

^1^ Values were presented as means ± pooled SEM, *n* = 6. ^2^ BL, multi-strain probiotics; PSE, *Perilla frutescens* seed extract; −PSE−BL, basal diet; +PSE−BL, basal diet + 1 g/kg *Perilla frutescens* seed extract; +BL−PSE, basal diet + 2 g/kg multi-strain probiotics; +BL + PSE, basal diet+2 g/kg multi-strain probiotics + 1 g/kg *Perilla frutescens* seed extract. ^A,B^ Different letters on the shoulders of peers indicate significant differences (*p* < 0.05).

**Table 7 animals-12-02246-t007:** Effects of BL and PSE on serum antioxidant indices of weaned piglets ^1^.

Items ^2^	PSE × BL				Pooled SEM	*p*-Value		
	−PSE	−PSE	+PSE	+PSE		PSE	BL	Interaction
	−BL	+BL	−BL	+BL				
SOD, U/mL								
Day 7	33.00	28.06	29.39	29.99	1.65	0.615	0.204	0.110
Day 14	41.28	41.35	37.72	40.31	1.36	0.105	0.337	0.363
Day 21	40.50	39.09	43.32	47.15	1.39	0.001	0.393	0.074
T-AOC, U/mL								
Day 7	0.96	0.99	0.94	0.96	0.03	0.492	0.458	0.818
Day 14	1.19	1.13	1.16	1.11	0.03	0.553	0.144	0.977
Day 21	1.01	1.06	1.05	1.10	0.03	0.183	0.127	0.936
CAT, U/mL								
Day 7	1.29	1.01	1.28	1.18	0.28	0.764	0.506	0.760
Day 14	2.00	1.88	1.31	3.09	0.67	0.701	0.231	0.174
Day 21	1.97	2.45	2.85	2.21	0.42	0.428	0.874	0.208
MDA, nmol/mL								
Day 7	6.43 ^A^	4.18 ^AB^	3.49 ^B^	5.15 ^AB^	0.77	0.011	0.095	0.046
Day 14	4.44	3.37	3.39	3.18	0.36	0.094	0.086	0.241
Day 21	6.26	5.18	5.03	5.52	0.49	0.377	0.562	0.128
GSH-Px, U/mL								
Day 7	702.23	717.85	683.92	845.08	44.27	0.233	0.060	0.116
Day 14	545.64	472.86	472.29	566.19	41.67	0.813	0.803	0.060
Day 21	527.64	520.25	478.45	444.24	48.77	0.214	0.674	0.786

^1^ Values were presented as means ± pooled SEM, *n* = 6. ^2^ BL, multi-strain probiotics; PSE, *Perilla frutescens* seed extract; −PSE−BL, basal diet; +PSE−BL, basal diet + 1 g/kg *Perilla frutescens* seed extract; +BL−PSE, basal diet + 2 g/kg multi-strain probiotics; +BL + PSE, basal diet+2 g/kg multi-strain probiotics + 1 g/kg *Perilla frutescens* seed extract. ^A,B^ Different letters on the shoulders of peers indicate significant differences (*p* < 0.05).

**Table 8 animals-12-02246-t008:** Effects of BL and PSE on antioxidant indices of liver and spleen of weaned piglets ^1^.

Items ^2^	PSE × BL				Pooled SEM	*p*-Value		
	−PSE	−PSE	+PSE	+PSE		PSE	BL	Interaction
	−BL	+BL	−BL	+BL				
Liver								
T-AOC, U/mg protein	2.77	4.52	5.15	6.79	0.65	0.002	0.016	0.936
MDA, nmol/mg protein	30.65	31.89	18.54	19.33	1.80	<0.001	0.580	0.901
Spleen								
T-AOC, U/mg protein	4.20	3.46	4.21	4.88	0.53	0.192	0.950	0.195
MDA, nmol/mg protein	45.21	30.89	25.20	18.64	3.02	<0.001	0.003	0.215

^1^ Values were presented as means ± pooled SEM, *n* = 6. ^2^ BL, multi-strain probiotics; PSE, *Perilla frutescens* seed extract; −PSE−BL, basal diet; +PSE−BL, basal diet + 1 g/kg *Perilla frutescens* seed extract; +BL−PSE, basal diet + 2 g/kg multi-strain probiotics; +BL + PSE, basal diet+2 g/kg multi-strain probiotics + 1 g/kg *Perilla frutescens* seed extract.

**Table 9 animals-12-02246-t009:** Effects of BL and PSE on intestinal antioxidant indices of weaned piglets ^1^.

Items ^2^	PSE × BL				Pooled SEM	*p*-Value		
	−PSE	−PSE	+PSE	+PSE		PSE	BL	Interaction
	−BL	+BL	−BL	+BL				
Duodenum								
T-AOC, U/mg protein	2.22	1.59	1.71	1.69	0.20	0.318	0.118	0.148
MDA, nmol/mg protein	9.82	12.79	10.70	11.40	1.42	0.857	0.208	0.431
Jejunum								
T-AOC, U/mg protein	15.45	15.66	15.32	16.13	1.05	0.871	0.632	0.777
MDA, nmol/mg protein	20.17 ^A^	11.61 ^BC^	10.85 ^C^	15.36 ^B^	1.14	0.024	0.090	<0.001
Ileum								
T-AOC, U/mg protein	1.10	1.46	0.97	1.71	0.21	0.773	0.019	0.396
MDA, nmol/mg protein	23.66	16.85	19.77	19.65	2.81	0.848	0.233	0.248

^1^ Values were presented as means ± SEM, *n* = 6. ^2^ BL, multi-strain probiotics; PSE, *Perilla frutescens* seed extract; −PSE−BL, basal diet; +PSE−BL, basal diet + 1 g/kg *Perilla frutescens* seed extract; +BL−PSE, basal diet + 2 g/kg multi-strain probiotics; +BL + PSE, basal diet+2 g/kg multi-strain probiotics + 1 g/kg *Perilla frutescens* seed extract. ^A,B,C^ Different letters on the shoulders of peers indicate significant differences (*p* < 0.05).

## Data Availability

The data are all contained and presented within the article.

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
