# Peer review of "Effects of Multi-Strain Probiotics and Perilla frutescens Seed Extract Supplementation Alone or Combined on Growth Performance, Antioxidant Indices, and Intestinal Health of Weaned Piglets"

_animals, 2022, doi:10.3390/ani12172246_

Round 1

Reviewer 1 Report

The study entitled “Effects of multi-probiotics and perilla seed extract supplementation alone or combined on growth performance, antioxidant indices, and intestinal health of weaned piglets” was designed to study the effect of multi-strain probiotics (BL) and perilla seed extract (PSE),
alone or in combination, on weaning piglet’s growth performance, antioxidant properties, and intestinal health of weaned piglets.

The manuscript cannot be accepted due to the following reasons:

1.      The manuscript was not numbered, therefore making it difficult to make reference.

2.      The experimental design of the study is not known. The authors stated 2 x 2 factorial design but this is not correct. 1. What are the 2 factors? 2. What are the 2 levels of each factor? If you are able to provide answers to these questions, that makes it 2 x 2 factorial arrangement.

3.      The presence of the multi-strain probiotics (BL) and perilla seed extract (PSE) were not tested in the feed.

4.      There was no effect of interaction in this study but authors consistently mentioned interaction throughout the manuscript. What authors referred to as interactions were simply addictive or combine effect of two products, thus BL + PSE not BL x PSE.

5.      The overall performance result in table 2 (D1-21), does not make sense. The difference between 0.617 vs 0.625 has P-value of 0.006, but the difference between 0.617 vs 0.637 has P-value of 0.034. Similarly, the difference between 0.617 vs 0.632 has P-value of 0.807.

6.      The foot note “Values were presented as means ± SEM, n = 6” is not true for tables 2, 3, 4, 6, 7, and 8.

7.      Authors should not start a sentence with abbreviations.

Author Response

  1. The manuscript was not numbered, therefore making it difficult to make reference.

We appreciate the comment from the reviewer. I am sorry to trouble you because we did not use the MS Word/LaTeX when submitted this manuscript, so there was no numbered line when you commented it. In the revised manuscript, we used MS Word/LaTeX.

  1. The experimental design of the study is not known. The authors stated 2 x 2 factorial design but this is not correct. 1. What are the 2 factors? 2. What are the 2 levels of each factor? If you are able to provide answers to these questions, that makes it 2 x 2 factorial arrangement.

Thanks for your kind comments. we are sorry to confuse you that we did not writer clearly about the experiment design. PSE has two levels:0 and 1g/kg, BL has 2 levels: 0 and 2g/kg. CON group = 0 PSE and 0 BL; PSE group = 1g/kg PSE +0BL

BL group = 2 g/kg BL + 0PSE; BL+PSE= 1g/kg PSE +2 g/kg BL. We have provided these in the revised manuscript in lines 79-80 and added mean of separate PSE effect and BL effect in every table. This kind of design some published paper writer as one-way ANOVA experimental design but also two-way ANOVA. We provide the same design using two-way AONVA below published papers.

González-Ortiz, G., Lee, S. A., Vienola, K., Raatikainen, K., Jurgens, G., Apajalahti, J., & Bedford, M. R. (2022). Interaction between xylanase and a proton pump inhibitor on broiler chicken performance and gut function. Animal Nutrition, 8(1), 277-288.  https://doi.org/10.1016/j.aninu.2021.06.005

Guo, F., Wang, F., Ma, H., Ren, Z., Yang, X., & Yang, X. (2021). Study on the interactive effect of deoxynivalenol and Clostridium perfringens on the jejunal health of broiler chickens. Poultry science, 100(3), 100807. https://doi.org/10.1016/j.psj.2020.10.061

Park, J. H., & Kim, I. H. (2015). Interactive effects of fenugreek (Trigonella foenum-graecum L.) seed extract supplementation and dietary metabolisable energy levels on the growth performance, total tract digestibility, blood profiles, and excreta gas emission in broiler chickens. Animal Production Science, 56(10), 1677-1682.

https://doi.org/10.1071/AN14834

  1. The presence of the multi-strain probiotics (BL) and perilla seed extract (PSE) were not tested in the feed.

   We appreciate your kind comments regarding our manuscript. We did not add the PSE or BL in the feed formulate. We firstly formulated feed for the control group followed by additional additives.

  1. There was no effect of interaction in this study but authors consistently mentioned interaction throughout the manuscript. What authors referred to as interactions were simply addictive or combine effect of two products, thus BL + PSE not BL x PSE.

We appreciate your comments on our manuscript. We revisited the results and found that there were no interactions on mucin2, or jejunal VH, CD and V/C. but there were really interactions on ileal VH, IL-1β, MDA concentration in the jejunum. We rewrote the results in lines 198, 204, and 213-214.

  1. The overall performance result in table 2 (D1-21), does not make sense. The difference between 0.617 vs 0.625 has P-value of 0.006, but the difference between 0.617 vs 0.637 has P-value of 0.034. Similarly, the difference between 0.617 vs 0.632 has P-value of 0.807.

Thanks for your comments. We have added the mean of BL effect and PSE effect in Table2.  0.598 vs 0.628 has P-value of 0.006, while 0.621 vs  0.635 has the P-value of 0.034. Because this is main effect P-value so we should provide main effect mean. We are sorry not to present data fully.

  1. The foot note “Values were presented as means ± SEM, n = 6” is not true for tables 2, 3, 4, 6, 7, and 8.

Thanks for your kind comment. Each treatment has 6 repeat pens. We chose one pig whose body weight close to the mean of body weight of each pen to euthanize. A lot of 24 data were run in the SAS procedure.

  1. Authors should not start a sentence with abbreviations.

     Thanks for your comments regarding our manuscript. We are sorry to make this low mistake. We have corrected this error in line 30, 33, 35, 83, 85, 160,168, 196, and 303.

Reviewer 2 Report

Dear Author

the suggest are in file

Author Response

Rev Effects of multi-probiotics and perilla seed extract supplementation alone or combined on growth performance, antioxidant indices, and intestinal health of weaned piglets
The author must to put number in lines.

Thanks for your comments. We are sorry that we don’t use MS Word/LaTeX asked by journal. In the revised manuscript, we provided number lines.

Simple summary:
Perilla write to cientific name

We appreciate your comments. We corrected the name of perilla. Actually, it is perilla frutescens seed extract. We have revised the name in the title, simple summary, and in the text in lines 2, 16, 19,22, 40, 56, and footnote in Table 2,3, 4, 5, 6,7, 8, and 9, and figure 1.

Abstract how probiotics?

Thank you very much. The probiotics contains Bacillus subtilis, Bacillus subtilis var natto, and Lactobacillus sporogenes. Because the Abstract had a limit of words within 200, so we don’t introduce the component in this part. We provide an introduction to probiotics in line 84.

Introduction
How bacteriocins?

Thanks for your comments. we added the bacteriocins in lines 45-46: Notably, the bacteriocins secreted by Bacillus and Lactobacillus without toxicity can lyse cell walls and consume bacterial proton movement, leading to cell death [2].

ADFI the author should to write to signficated.

Thank you very much. We give the full name of ADFI in line 61 since it is the first time that it appears. We think the ADFI is not significant in the manuscript and the only one related research on livestock was reported. Therefore we don’t give detailed descriptions about ADFI.
Methods
How month of the year? Winter? Springer?

Thanks for the comments. The animal experiment was conducted from 2021.7.12 to 2021.8.02. It was summer. We controlled the temperature using insulation lamps and fans.  
In text - The liver samples were taken from the largest leaf after removing the surface
membrane and stored at - 80 °C. Spleen tissues were frozen at - 80 °C for antioxidant analysis. Individual chyme collected from the ileum was stored in sterile Eppendorf tubes and rapidly stored at - 80 °C rapidly until analysis of microbiota- how the reference??

Thank you very much. In terms of collecting liver and spleen samples, we want to keep the same site taken for each sample, for liver, we chose the largest leaf, therefore, we did not refer to any references. The collection and assay of microbiota reference the method of Hu et al. (2017). We added the reference in lines 106-107.

[5] Hu, L., Peng, X., Chen, H., Yan, C., Liu, Y., Xu, Q., ... & Che, L. (2017). Effects of intrauterine growth retardation and Bacillus subtilis PB6 supplementation on growth performance, intestinal development and immune function of piglets during the suckling period. European Journal of Nutrition, 56(4), 1753-1765. https://doi.org/10.1007/s00394-016-1223-z

2.3. Assay of Oxidant and Antioxidant Indices in Serum, Liver, and Spleen Tissues, and Small Intestine Mucosa, The text does not have the reference. It is better put it

Thanks for the comments. We added the reference[22] in line 109.

[22] Chen, J.; Yu, B.; Chen, D.; Huang, Z.; Mao, X.; Zheng, P.; Yu, J.; Luo, J.; He, J. Chlorogenic acid improves intestinal barrier functions by suppressing mucosa inflammation and improving antioxidant capacity in weaned pigs. The Journal of Nutritional Biochemistry. 2018, 59, 84-92. https://doi.org/10.1016/j.jnutbio.2018.06.005

2.4. Small Intestine Biochemical Analysis. The text does not have the reference. It is better put it. It better to put in text how the function to do the biochemical analysis.

Thanks for the comments. We added the reference [24] in line 120. We added the sentence “why we assay the biochemical index”. Line 119-120: As intestinal morphology plays a vital function in absorbing nutrients and changes after weaning [23], we evaluate the morphology of duodenum, jejunum, and ileum.  

[24] Wan, J.; Zhang, J.; Chen, D.; Yu, B.; Huang, Z.; Mao, X.; Zheng, P.; Yu, J.; He, J. Alginate oligosaccharide enhances intestinal integrity of weaned pigs through altering intestinal inflammatory responses and antioxidant status.2018. RSC Adv, 8, 13482-13492.

[23] Hampson, D. J.1986. Alterations in piglet small intestinal structure at weaning. Res. Vet. Sci. 40, 32-40. doi: 10.1016/S0034-5288(18)30482-X

2.5. Total RNA Extraction and Quantitative Real-Time PCRThe text does not have the reference. It is better put it. It better to put in text how the function to do the total RNA extraction

Thank you very much. According to your suggestions, we supplemented reference and this sentence in lines 135-136: To assess the effects of BL and PSE on the gene expression of gut barrier genes and inflammatory factors, we measured the gut mucosal genes according to the method of Hu et al. [5].

[5] Hu, L., Peng, X., Chen, H., Yan, C., Liu, Y., Xu, Q., ... & Che, L. (2017). Effects of intrauterine growth retardation and Bacillus subtilis PB6 supplementation on growth performance, intestinal development and immune function of piglets during the suckling period. European Journal of Nutrition, 56(4), 1753-1765. https://doi.org/10.1007/s00394-016-1223-z

2.6 - Microbial DNA Isolation and RT-qPCR.It better to put in text how the function to do microbial DNA isolation. What the relevance in the paper?

  Thanks for your kind suggestions. We give the reference and the relevance in lines 148-149: Gut microbiota are easily affected by dietary elements, especially, weaned piglets [26]. Therefore, the effects of BL and PSE on microorganisms of weaning pigs were assessed based on the previous description [27].

[26] Zhang, D.; Ji, H.; Liu, H.; Wang, S.; Wang, J.; Wang, Y. Changes in the
 diversity and composition of gut microbiota of weaned piglets after oral
 administration of Lactobacillus or an antibiotic. Appl. Microbiol. Biot. 2016, 100, 10081-10093. https://doi.org/10.1007/s00253-016-7845-5

[27] Chen, H.; Mao, X.; He, J.; Yu, B.; Huang, Z.; Yu, J.; Zheng, P.; Chen, D. Dietary fibre affects intestinal mucosal barrier function and regulates intestinal bacteria in weaning piglets. Brit. J. Nutr. 2013, 110, 1837-1848. https://doi.org/10.1017/S000711451300129310081-10093. https://doi.org/10.1007/s00253-016-7845-5

Table 2 BW- what the significance? D? day? It is better to put the significance

Thank you very much. We are sorry to confuse you because the tables were unreasonable design. In the original manuscript, we did not give the main effect mean of PSE or BL. In the revised manuscript, we give superscripts A and B to mark the significance and give the main effect mean in every table. There is no significance in BW but in G/F in Table 2

  1. Discussion
    4.1.Growth Performance
    In text ...... This result is in agreement with similar studies of Bacillus and Lactobacillus probiotics in weaned piglets [7,25,26] and with the observation that Bacillus spp.-based probiotics stimulate the activity of non-starch polysaccharide enzymes and proteases and augment nutrient digestibilit. It is necessary to put a significance. Why it stimulate the activity the enzymes. Can not only write

Thank you very much. We supplemented the measurement of digestive enzyme indicators in Table 5. PSE elevated the amylase activity. Dietary BL induced higher trypsin activity. We rewrite this in the discussion.

Lines 224-226: Bacillus spp.-based probiotics stimulate the activity of non-starch polysaccharide enzymes and proteases and augment nutrient digestibility [3]. Therefore, we guessed that the improved trypsin activity by BL may stimulate the decomposition of proteins and then increase nutrients absorption.

Lines 230-231: In the present study, improved amylase activity by PSE was more conducive to decomposing starch to easily absorbed maltose and glucose

4.2. Diarrhea Scores and Microbes
due to the complex internal environment to write the possible complex environment.

 We appreciate your questions. We rethink the discussion about “PSE had no effect on E. coli” and deleted the possible complex environment. We give the new ideas to discuss.

Lines 243-247: However, the latest report revealed that perilla extract had no inhibition on E. coli or Salmonella in vitro [39]. On the one hand, perilla varieties, harvest season, and planting environment will have impacts on its composition [39]. Another hand, when entering the body, PSE may be digested by digestive enzymes and absorbed by intestinal epithelial cells, so they cannot play the roles in the hindgut.

4.3. Gut Barrier Gene Expression
Phenolic acid and flavonoid constituents have a protective function in regulating expression levels of ZO-1 and occludin in weaning pigs and rats challenged by lipopolysaccharide [43,44].Why?? Explain

We really appreciate your comments regarding this manuscript. We give the explanation in lines 265-267: Phenolic acid and flavonoid constituents have a protective function on expression levels of ZO-1 and occluding because it is a potent inhibitor of protein tyrosine kinases that prevents tight junction protein from being phosphorylated [47]

4.5 Interactive Effect
Zangeronimo et al. [54] certified that higher doses. How much about the doses? 1000mg

Thank you very much. The dose is 1000 mg/kg = 1g/kg. we give the dose in lines 293-294: Analogously, Zangeronimo et al. [57] certified that 1 g/kg plant extract mixed with 3 g/kg probiotics in the diet of weaned piglets decreased the VH compared with low doses.